# AGENTIC AI FOR SCIENTIFIC DISCOVERY: A SURVEY OF PROGRESS, CHALLENGES, AND FUTURE DIRECTIONS

**Mourad Gridach, Jay Nanavati, Khaldoun Zine El Abidine, Lenon Mendes & Christina Mack**
Real World Solution, Applied AI Science, Cambridge, United Kingdom
`{firstname.lastname}@iqvia.com`

## ABSTRACT

The integration of Agentic AI into scientific discovery marks a new frontier in research automation. These AI systems, capable of reasoning, planning, and autonomous decision-making, are transforming how scientists perform literature review, generate hypotheses, conduct experiments, and analyze results. This survey provides a comprehensive overview of Agentic AI for scientific discovery, categorizing existing systems and tools, and highlighting recent progress across fields such as chemistry, biology, and materials science. We discuss key evaluation metrics, implementation frameworks, and commonly used datasets to offer a detailed understanding of the current state of the field. Finally, we address critical challenges, such as literature review automation, system reliability, and ethical concerns, while outlining future research directions that emphasize human-AI collaboration and enhanced system calibration.

## 1 INTRODUCTION

The rapid advancements of Large Language Models (LLMs) (Touvron et al., 2023; Anil et al., 2023; Achiam et al., 2023) have opened a new era in scientific discovery, with Agentic AI systems (Kim et al., 2024; Guo et al., 2023; Wang et al., 2024; Abramovich et al., 2024) emerging as powerful tools for automating complex research workflows. Unlike traditional AI, Agentic AI systems are designed to operate with a high degree of autonomy, allowing them to independently perform tasks such as hypothesis generation, literature review, experimental design, and data analysis. These systems have the potential to significantly accelerate scientific research, reduce costs, and expand access to advanced tools across various fields, including chemistry, biology, and materials science.

Recent efforts have demonstrated the potential of LLM-driven agents in supporting researchers with tasks such as literature reviews, experimentation, and report writing. Prominent frameworks, including LitSearch (Ajith et al., 2024), ResearchArena (Kang & Xiong, 2024), SciLitLLM (Li et al., 2024c), CiteME (Press et al., 2024), ResearchAgent (Baek et al., 2024) and Agent Laboratory (Schmidgall et al., 2025), have made strides in automating general research workflows, such as citation management, document discovery, and academic survey generation. However, these systems often lack the domain-specific focus and compliance-driven rigor essential for fields like biomedical domain, where the structured assessment of literature is critical for evidence synthesis. For example, Agent Laboratory demonstrated high success rates in data preparation, experimentation, and report writing. However, its performance dropped significantly in the literature review phase, reflecting the inherent challenges of automating structured literature reviews. Moreover, questions about system reliability, reproducibility, and ethical governance continue to pose significant hurdles.

This survey aims to provide a comprehensive review of Agentic AI for scientific discovery. We categorize existing systems into autonomous and collaborative frameworks, detailing the datasets, implementation tools, and evaluation metrics that support these innovations. By highlighting the current state of the field and discussing open challenges, we hope to inspire further research and development in Agentic AI, ultimately encouraging more reliable and impactful scientific contributions.

## 2 Agentic AI: Foundations and Key Concepts

### 2.1 Definition and Characteristics

The concept of an "agent" has a rich history and has been explored across various disciplines including philosophy. It has been discussed by influential philosophers starting from Aristotle to Hume among others. Generally, an "agent" is an entity that has the ability to act, while the concept of "agency" refers to the exercise or representation of this ability Schlosser (2019). In artificial intelligence, an agent is an autonomous intelligent entity capable of performing appropriate and contextually relevant actions in response to sensory input, whether operating in physical, virtual, or mixed-reality environments. Agentic AI introduces a new paradigm in the AI community, highlighting the concept of embodied intelligence and showing the importance of an integrated framework for interactive agents within complex systems Huang et al. (2024c). This paradigm stems from the understanding that intelligence emerges from the intricate interaction between key processes such as autonomy, learning, memory, perception, planning, decision-making and action.

### 2.2 Single Agent vs. Multi-Agents

With the explosion of both research papers and industrial applications of agentic AI, a new debate emerged on whether single or multi-agent systems are best suited for solving complex tasks. In general, single agent architectures shine when dealing with well-defined problems and feedback from the user is not needed, while multi-agent architectures are suitable for solving problems that involve collaboration and multiple runs are needed.

**Single Agent** In nutshell, a single agent is able to achieve its goal independently without relying on assistance or feedback from other AI agents, even if multiple agents coexist within the same environment. However, there may be opportunities for humans to be in the loop by providing feedback for agent guidance. More specifically, a single agent with an LLM backbone capable of handling multiple tasks and domains is called LM-based agent. It is able to perform reasoning, planning and tool execution on their own. Given an input prompt, an agent uses the tool to execute its task. Common applications using a single agent include Scientific Discovery Lu et al. (2024); Ghafarollahi & Buehler (2024a); Kang & Xiong (2024); Xin et al. (2024), web scenarios Nakano et al. (2021); Deng et al. (2024); Furuta et al. (2024); Zhou et al. (2024), gaming environments Yuan et al. (2023); Nottingham et al. (2023), and healthcare Zhang et al. (2023); Abbasian et al. (2023).

**Multi-agents** These architectures involve two or more agents in interactions between each other. Originally inspired by Minsky's Society of Mind Minsky (1988) where he introduced a novel theory of intelligence based on the interactions between smaller agents with specific functions leading to intelligence. Multi-agents require a careful interoperability among various agents, specifically in their communications and information sharing. Multi-agent systems are a powerful collaborative framework when dealing with problems involving tasks that spans multiple domains where each agent is expert in a particular domain. In NLP, each agent can use the same or different LLM backbone. In contrast, agents may use the same tools or distinct ones, with each agent typically embodying a unique persona. Multi-agent systems are widely explored in domains including scientific discovery Schmidgall et al. (2025); Baek et al. (2024); Ghafarollahi & Buehler (2024b); Swanson et al. (2024); Xiao et al. (2024), software development Qian et al. (2024); White (2024) and healthcare Tang et al. (2024); Kim et al. (2024). While multi-agent systems are powerful in solving difficult problems in complex environments, the communication and interaction between agents remain one of the challenges compared to single agent systems.

## 3 Taxonomy of Agentic AI for Scientific Discovery

The scope of Agentic AI for scientific discovery is vast, encompassing tasks such as hypothesis generation, experiment design, data analysis, and literature review. By automating these traditionally labor-intensive processes, Agentic AI has the potential to accelerate the pace of scientific discovery, reduce costs, and democratize access to advanced research tools. However, the true power of Agentic AI lies in its ability to augment human expertise rather than replace it. These systems are increasingly being designed to collaborate with researchers, providing insights, generating novel ideas, and handling repetitive tasks, thereby freeing up scientists to focus on creative and high-level

problem-solving. As the field continues to evolve, its applications in scientific discovery are expanding across diverse domains, from chemistry and biology to materials science and healthcare. Agentic AI systems can be broadly categorized based on their level of autonomy, interaction with researchers, and scope of application.

## 3.1 FULLY AUTONOMOUS SYSTEMS

Fully autonomous systems are designed to operate independently, automating end-to-end scientific workflows with minimal human intervention. These systems leverage advanced AI capabilities, such as natural language understanding, planning, and decision-making, to perform complex tasks ranging from hypothesis generation to experiment execution. Boiko et al. (2023) developed Co-scientist, an autonomous AI agent powered by GPT-4 that plans, designs, and executes chemical experiments. Similarly, M. Bran et al. (2024) introduced ChemCrow, which extends the capabilities of GPT-4 by integrating 18 expert-designed tools for tasks such as organic synthesis, drug discovery, and materials design. It demonstrates the potential of fully autonomous systems to tackle complex, domain-specific challenges. ProtAgents Ghafarollahi & Buehler (2024a) was proposed for protein design and molecular modeling. It leverages LLMs and reinforcement learning to optimize protein structures, predict folding patterns, and perform molecular docking simulations. ProtAgents can autonomously generate, test, and refine protein sequences to meet desired biochemical properties. LLaMP (Large Language Model for Materials Prediction) Chiang et al. (2024) is an autonomous AI agent for materials science, using RAG to predict material properties and optimize formulations. It autonomously conducts atomic simulations and materials discovery, aiding applications in nanotechnology, energy storage, and catalysis. The main advantage of these systems is their efficiency in environments where tasks are well-defined, repetitive, or require high precision. They can significantly accelerate research by automating time-consuming processes. However, they may struggle with tasks that require creativity, domain-specific intuition, or interdisciplinary knowledge, highlighting the need for human oversight in certain scenarios.

## 3.2 HUMAN-AI COLLABORATIVE SYSTEMS

Human-AI collaborative systems emphasize the synergy between AI and researchers, combining the computational power of AI with the creativity and human expertise. Swanson et al. (2024) proposed Virtual Lab, an AI-human collaborative framework that conducts interdisciplinary scientific research. It organizes team meetings and individual tasks to solve complex problems, such as designing nanobody binders for SARS-CoV-2. ODonoghue et al. (2023) developed BioPlanner, an AI-driven research planning tool that designs experimental protocols by converting scientific goals into pseudocode-like steps. It assists researchers in structuring wet-lab experiments efficiently but does not conduct them autonomously. Also, Prince et al. (2024) introduced CALMS (Context-Aware Language Model for Science), an AI-powered lab assistant that interacts with scientists and laboratory instruments. It provides real-time contextual assistance in experiments, helping with procedure guidance, data interpretation, and workflow optimization, though it does not autonomously execute experiments. More recently, Schmidgall et al. (2025) introduced Agent Laboratory, a framework that accepts human-provided research ideas and autonomously progresses through literature review, experimentation, and report writing. The advantages of these AI-driven scientific frameworks lie in their ability to accelerate research, enhance experimental design, and optimize decision-making in fields like genetics, materials science, and chemistry. However, their limitations stem from their reliance on human oversight, data quality, and interpretability. Therefore, they still require manual validation and execution.

## 4 AGENTIC AI FOR LITERATURE REVIEW

Scientific discovery is an iterative process that builds upon existing knowledge, requiring researchers to systematically explore and synthesize prior work. A literature review serves as the foundation for this process, enabling scientists to identify key trends, evaluate methodologies, and recognize gaps in knowledge that can drive new research directions. In fields such as chemistry, biology, materials science, healthcare, and artificial intelligence, a well-conducted literature review is essential for framing research questions, selecting appropriate experimental or computational approaches, and ensuring reproducibility. With the exponential growth of scientific publications, traditional man-

ual reviews have become increasingly challenging. Researchers now rely on advanced tools such as autonomous agents, to navigate vast datasets of scientific literature efficiently. These technologies facilitate automatic extraction of relevant information, trend analysis, and predictive modeling, accelerating the rate of discovery.

Agentic AI systems have the potential to address these challenges by automating information retrieval, extraction, and synthesis. However, automating literature review is a complex task that requires advanced natural language understanding, domain-specific knowledge, and the ability to handle ambiguity and nuance. Several frameworks have been developed to automate or augment the literature review process using Agentic AI. SciLitLLM Li et al. (2024c) is a proposed framework designed to enhance the scientific literature understanding. It employs a hybrid strategy that combines continual pre-training (CPT) and supervised fine-tuning (SFT) to infuse domain-specific knowledge and improve instruction-following abilities. While SciLitLLM demonstrates improved performance on tasks such as document classification, summarization, and question answering, making it a valuable tool for literature review, the framework relies heavily on high-quality training data, which may not always be available for emerging fields. Ajith et al. (2024) introduced LitSearch, a benchmark designed to evaluate retrieval systems on complex literature search queries in machine learning and NLP. The main strengths of LitSearch is its ability to provide a standardized framework for assessing the performance of retrieval systems, enabling researchers to compare different approaches and identify areas for improvement. In contrary, the benchmark is limited to specific domains (ML and NLP), which may restrict its applicability to other fields. ResearchArena Kang & Xiong (2024) is a benchmark for evaluating LLM-based agents in academic surveys, dividing the process into three stages: information discovery, selection, and organization. It helps assess AI performance in structured literature reviews but struggles to capture the complexity of real-world reviews. CiteME Press et al. (2024) evaluates language models' ability to accurately attribute scientific claims to their sources, focusing on machine learning literature. While CiteME addresses a crucial aspect of literature review by ensuring accurate citation, it is limited in scope, restricting its application to other fields.

Despite the progress made by existing frameworks, several challenges remain in automating the literature review process. While frameworks such as SciLitLLM and ResearchArena demonstrate promising results, they often struggle with tasks requiring deep domain-specific knowledge and nuanced understanding. This limitation is further highlighted in Agent Laboratory Schmidgall et al. (2025), where a significant performance drop was observed during the literature review phase, emphasizing the complexity of automating this process. Another challenge lies in human-AI collaboration, as many existing frameworks prioritize fully autonomous workflows. This approach may limit usability for researchers who want to explore their unique ideas, underscoring the need for collaborative approaches that effectively integrate human expertise with AI capabilities. Generalizability is also a major obstacle, as many frameworks are designed for specific domains like machine learning, chemistry, or materials science, which restricts their application in other fields.

## 5 AGENTIC AI FOR SCIENTIFIC DISCOVERY

Agentic AI systems are revolutionizing the scientific research process by automating and augmenting various stages of the research lifecycle, from ideation and experimentation to paper writing and dissemination. Figure 1 depicts the agentic AI workflow for scientific discovery. These systems leverage the capabilities of LLMs and other AI technologies to streamline workflows, reduce human effort, and accelerate the pace of discovery. In this section, we explore how Agentic AI is transforming scientific discovery, supported by case studies and a discussion of key challenges. The research lifecycle traditionally involves several stages, including problem identification, literature review, hypothesis generation, experiment design, data analysis, and publication. Agentic AI systems are being deployed to automate or augment each of these stages, enabling researchers to focus on high-level decision-making and creative problem-solving. Here are the main steps:

- **Ideation:** Ideation refers to the process of generating, refining, and selecting research ideas or hypotheses. AI agents automate this process by analyzing existing literature, identifying gaps, and proposing novel hypotheses, thereby accelerating the initial stages of research Baek et al. (2024).

- **Experiment design and execution:** Experiment design involves planning and structuring experiments to test hypotheses, while execution refers to carrying out these experiments. AI agents autonomously design and execute complex experiments by integrating tools for planning, optimization, and robotic automation Boiko et al. (2023).

- **Data analysis and interpretation:** Data analysis involves analyzing experimental data to extract meaningful insights, while interpretation refers to drawing conclusions and identifying patterns. Agents can process large datasets and generate insights that might be overlooked by researchers, enhancing the accuracy and efficiency of this stage.

- **Paper writing and dissemination:** Paper writing involves synthesizing research findings into a coherent and structured manuscript, while dissemination refers to sharing the research with the scientific community through publications or presentations. AI agents automate the writing of research papers, ensuring clarity, coherence, and adherence to academic standards, thereby reducing the time and effort required for publication Lu et al. (2024).

By relying on LLM-augmented agents, these systems have made a significant strides in scientific discovery in domains such as chemistry, biology, materials science as well as general science where the main dream is to develop a fully autonomous AI scientist. In chemistry, AI agents are transforming key areas such as molecular discovery and design, reaction prediction, and synthesis planning by accelerating the identification of novel compounds and optimizing synthetic routes. Additionally, they contribute to laboratory automation, integrating with robotic systems to execute experiments autonomously, and enhance computational chemistry by running molecular simulations for reaction kinetics and thermodynamics.

Coscientist Boiko et al. (2023) is an autonomous AI agent powered by GPT-4 that plans, designs, and executes chemical experiments. It integrates modules for web search, documentation analysis, code execution, and robotic automation, enabling it to handle multi-step problem-solving and data-driven decision-making. For example, Coscientist successfully designed and optimized a palladium-catalyzed cross-coupling reaction, demonstrating its potential to accelerate chemical discovery. Similarly, Ruan et al. (2024) proposed LLM-RDF (Large Language Model Reaction Development Framework), a framework that automates chemical synthesis using six LLM-based agents for tasks like literature search, experimental design, reaction optimization, and data analysis. Tested on copper/TEMPO catalyzed aerobic alcohol oxidation, it demonstrates end-to-end synthesis automation. It simplifies reaction development, making it more accessible to chemists without coding expertise. In the same context, Chiang et al. (2024) developed a novel framework called LLaMP (Large Language Model Made Powerful), designed for scientific discovery in chemistry by integrating RAG with hierarchical reasoning agents. It significantly reduces hallucination in material informatics by grounding predictions in high-fidelity datasets from the Materials Project (MP) and running atomistic simulations. LLaMP successfully retrieves and predicts key material properties such as bulk modulus, formation energy, and electronic bandgap, outperforming standard LLMs. Darvish et al. (2024) introduced Organa, an assistive robotic system designed for automating diverse chemistry experiments, including solubility screening, pH measurement, recrystallization, and electrochemistry characterization. Using LLMs for reasoning and planning, Organa interacts with chemists in natural language to derive experiment goals and execute multi-step tasks with parallel execution capabilities. In electrochemistry, it demonstrated the automation of complex processes, such as electrode polishing and redox potential measurement, achieving results comparable to human chemists while reducing execution time by over 20%. This system enhances scientific discovery by improving the reproducibility and efficiency of chemistry experiments. Other notable frameworks based on AI agents for scientific discovery in chemistry include ChatMOF Kang & Kim (2023), ChemCrow M. Bran et al. (2024) and MOOSE-CHEM Yang et al. (2024) among others.

In addition to chemistry, AI agents, powered by LLMs and multi-agent systems, are transforming biology by enabling automated data analysis, hypothesis generation, and experimental planning. These agents can extract insights from vast amounts of biological data, such as genomic sequences, protein structures, and biomedical literature, to accelerate research across fields like genetics, drug discovery, and synthetic biology. With capabilities such as gene-editing design, protein engineering, and systems biology modeling, AI agents are playing a critical role in scientific discovery for biology. By integrating with laboratory tools and robotic systems, they not only reduce human effort but also enhance research accuracy and reproducibility, bringing us closer to breakthroughs in personalized medicine, disease modeling, and bioinformatics. Xin et al. (2024) introduced BIA

(BioInformatics Agent), an AI agent leveraging LLMs to streamline bioinformatics workflows, particularly focusing on single-cell RNA sequencing (scRNA-seq) data analysis. BIA automates complex tasks like data retrieval, metadata extraction, and workflow generation, significantly improving bioinformatics research efficiency. It features a chat-based interface for designing experimental protocols, invoking bioinformatics tools, and generating comprehensive analytical reports without coding. BIA's innovative use of static and dynamic workflow adaptation allows it to refine bioinformatics analyses iteratively, demonstrating its potential to reduce the cognitive load on researchers and enhance scientific discovery in genomics and transcriptomics. Similarly, Xiao et al. (2024) developed CellAgent, an LLM-driven multi-agent system designed to automate single-cell RNA sequencing data analysis. It features three expert roles: Planner, Executor, and Evaluator, which collaborate to plan, execute, and evaluate data analysis tasks such as batch correction, cell type annotation, and trajectory inference. CellAgent reduces human intervention by incorporating a self-iterative optimization mechanism, achieving a 92% task completion rate and outperforming other scRNA-seq tools in accuracy and reliability. This framework significantly enhances biological research efficiency, making scRNA-seq analysis accessible to non-experts and enabling new biological discoveries. Liu et al. (2024) developed TAIS ( Team of AI-made Scientists), a semi-autonomous AI assistant for genetic research, designed to suggest and refine biological experiments using self-learning mechanisms. It helps scientists with data analysis, hypothesis generation, and experiment planning, but requires human validation before execution. Other works include ProtAgents Ghafarollahi & Buehler (2024a), AI Scientists Gao et al. (2024) and CRISPR-GPT Huang et al. (2024a).

In addition to the previous domains, AI agents are widely explored in other fields such as materials science Ni et al. (2024); Maqsood et al. (2024); Papadimitriou et al. (2024); Strieth-Kalthoff et al. (2024); Merchant et al. (2023); Kumbhar et al. (2025), general science Taylor et al. (2022); Yang et al. (2023b); Baek et al. (2024); Lu et al. (2024); Swanson et al. (2024); Qi et al. (2023); Ghafarollahi & Buehler (2024b); Schmidgall et al. (2025) as well as machine learning Li et al. (2024b); Huang et al. (2024b); Chan et al. among others. Figure 2 shows the summary of these agentic AI frameworks for scientific discovery in various domains.

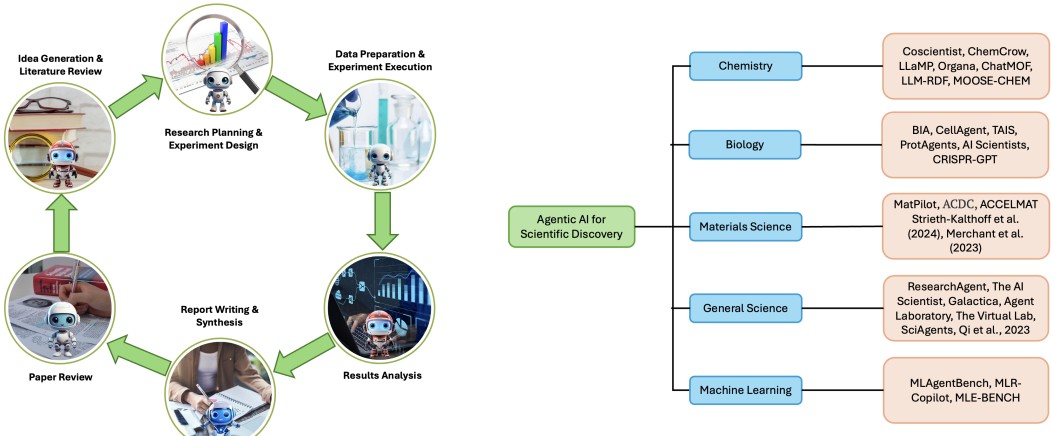

Figure 1: Agentic AI workflow for scientific discovery.

Figure 2: AI Agents frameworks for scientific discovery.

## 6 IMPLEMENTATION TOOLS, DATASETS AND METRICS

The development and evaluation of agentic AI systems for scientific discovery rely on a robust tools, curated datasets, and well-defined evaluation metrics. This section provides an overview of the key resources used in the field to facilitate the design, training, and assessment of autonomous AI agents for scientific discovery.

## 6.1 IMPLEMENTATION TOOLS

Agentic AI systems leverage a combination of foundational models, computational frameworks, and domain-specific tools to execute scientific tasks effectively. These tools can handle the creation of single and multi-agents frameworks. AutoGen is a comprehensive framework for managing multi-agent systems Wu et al. (2023). It is centered around the idea of "customizable and conversable agents.". It allows developers to define or program agents using both natural language and code, making it versatile for applications ranging from technical fields like coding and mathematics to consumer-oriented sectors such as entertainment. MetaGPT Hong et al. (2024), an intelligent agentic framework, streamlines the software development process. It emphasizes embedding human workflow processes into the task of LM agents and using an assembly line method to assign particular roles to different agents. Letta [1], an open-source agentic framework, allows the easy build and deployment of persistent agents as services. Letta is mainly based on the recent MemGPT paper Packer et al. (2023) and stands out as the framework explicitly incorporating cognitive architecture principles. Other impactful tools include CAMEL Li et al. (2023), LangChain, and AutoGPT Yang et al. (2023a).

## 6.2 DATASETS

Table 1 summarizes the commonly used datasets for agentic AI for scientific discovery. For scientific reasoning and discovery, most datasets are designed to evaluate the reasoning, planning, and collaborative capabilities of multiple AI agents in tasks like hypothesis generation, literature analysis, and experimental planning. In biology and chemistry, datasets such as LAB-Bench Laurent et al. (2024) and MoleculeNet Wu et al. (2018) are used to benchmark agents' ability to understand and analyze complex biological and chemical data. However, in emerging areas like materials discovery and entire research process automation, there is still a need for comprehensive benchmarks that assess the agents' real-world impact and adaptability. The development of such benchmarks would greatly enhance the evaluation of agentic AI systems, helping researchers gauge their applicability in complex and dynamic fields like genomics, drug discovery, and synthetic biology.

Table 1: Datasets and Benchmarks for Agentic AI for Scientific Discovery.

| Dataset/Benchmark | Domain | Purpose |
|---|---|---|
| **LAB-Bench** Laurent et al. (2024) | Biology | Evaluate reasoning and planning for biological research |
| **MoleculeNet** Wu et al. (2018) | Chemistry | Molecular property prediction |
| **ZINC Database** Irwin et al. (2012) | Chemistry | Virtual screening for drug discovery |
| **MatText** Alampara et al. (2024) | Materials Science | Text-based material property prediction |
| **MatSci-NLP** Song et al. (2023) | Materials Science | Language processing for materials science |
| **MaScQA** Zaki et al. (2024) | Materials Science | QA for materials science |
| **ChEMBL** Gaulton et al. (2012) | Chemistry | Bioactive molecule prediction |
| **PubChem** Kim et al. (2016) | Chemistry | Molecular feature extraction |
| **Mol-Instructions** Fang et al. (2023) | Biology/Chemistry | Protein and biomolecular-related tasks |
| **MPcules** Spotte-Smith et al. (2023) | Materials Science | Molecular properties |
| **AlphaFold Protein Structure** Varadi et al. (2022) | Biology | Protein structure prediction |
| **ICLR 2022 OpenReview** Lu et al. (2024) | Scientific Research | Performance evaluation of the automated paper reviewer |

---

[1]https://www.letta.com/

### 6.3 METRICS

Metrics in this field are diverse, depending on the specific task and domain. For reasoning and planning, metrics typically assess accuracy, task completion rates, and response coherence. In experimental prediction and scientific discovery, metrics like precision, recall, and prediction error are used to evaluate the quality and reliability of AI-generated results. Explainability and human evaluation also play a critical role in assessing how well these systems align with scientific goals. The recent proposed framework, Agent Laboratory Schmidgall et al. (2025), introduces additional evaluation metrics that provide a more comprehensive assessment of agentic AI systems. These include NeurIPS-style paper evaluation metrics such as quality, significance, clarity, soundness, presentation, and contribution, which are used to assess the scientific output of AI-generated research papers. Success rates track the percentage of successfully completed workflows, while human and automated reviewer comparisons ensure consistency and reliability in evaluations. Usability and satisfaction metrics, such as utility, continuation, and user satisfaction, are employed to assess the system's ease of use and overall user experience. However, for more complex tasks, such as multi-agent cooperation in experimental automation and hypothesis generation, standardized evaluation metrics are still in development. Establishing comprehensive metrics that combine objective performance measures (e.g., success rates and prediction accuracy) with subjective human assessments (e.g., user satisfaction and explainability) will be essential to accurately gauge the performance of these systems in real-world applications.

## 7 CHALLENGES AND OPEN PROBLEMS

While agentic AI systems hold immense promise for transforming scientific discovery, they also face significant challenges that must be addressed to realize their full potential. In this section, we discuss the main challenges facing the field of agentic AI for scientific discovery.

### 7.1 TRUSTWORTHINESS

Current research emphasizes avoiding overfitting to reflect real-world conditions, enhancing AI agent predictability Kapoor et al. (2024). The focus on agentic assurance and trustworthiness of AI agents for scientific discovery includes robust benchmarking practices to ensure the reliability and effectiveness of AI agents in real-world applications. It highlights the need for cost-controlled evaluations and the joint optimization of performance metrics such as accuracy, cost, speed, throughput, and reliability (e.g., task failure rates, recovery upon failure). This approach aims to develop efficient, practical AI agents for real-world deployment, avoiding overly complex and costly designs. Ongoing efforts also focus on improving the explainability and safety of AI agents, ensuring their actions and decisions can be understood and scrutinized by humans. This involves developing methods to make AI behavior more interpretable and provide clear explanations for their decisions. Research highlights the importance of avoiding overfitting and ensuring that benchmarks are designed to reflect real-world conditions, thus enhancing the practical utility of AI agents Li et al. (2024a); Aliferis & Simon (2024). These efforts stress the need for robust evaluation frameworks to maintain high generalization performance. Additionally, innovative methods to detect and prevent overfitting further contribute to the reliability and trustworthiness of AI systems. These studies collectively underscore the necessity of comprehensive evaluation practices to develop AI agents that are both accurate and dependable in real-world scenarios.

### 7.2 ETHICAL AND PRACTICAL CONSIDERATIONS

Ethical considerations and principles are a major focus for many research groups in both academia and industry. Ethics play a critical role in the development and deployment of AI agents, especially in critical domains such as healthcare. Managing bias is a key ethical risk, in addition to other matters of privacy, accountability, and compliance previously addressed. Therefore, there is an urgent need for transparency, accountability, and fairness in designing AI agents, and the need to prioritize these values throughout the development lifecycle. When incorporating LLMs into autonomous agents, ethical challenges become even more pronounced. LLMs, by nature, can amplify existing biases in training data, potentially leading to unethical or harmful outputs. They also pose risks in generating misleading, fabricated, or contextually inappropriate responses (hallucina-

tions), particularly detrimental in critical domains like healthcare. Agent-specific challenges further intensify these ethical considerations. In the future, autonomous agents may often operate collaboratively in decentralized environments or with tool-calling capabilities, such as automating financial transactions or managing sensitive health records. If one agent in a multi-agent system behaves unethically—whether due to adversarial tampering, incomplete ethical alignment, or systemic bias—it can compromise the integrity of the entire system. Addressing these issues requires robust oversight mechanisms, human-in-the-loop architectures, and frameworks to evaluate and mitigate these risks during training and deployment. Algorithms tackling bias detection and mitigation, such as adversarial debiasing Lim et al. (2023) and reweighting Zhu et al. (2021), can be incorporated into the training process to minimize the risk of perpetuating existing biases, enabling the detection and correction of biases in both data and model outputs.

### 7.3 POTENTIAL RISKS

Agentic AI offers exciting possibilities in scientific discovery but also introduces significant risks. As these systems take on complex tasks—such as data analysis, hypothesis generation, and experiment execution—data reliability and bias become major concerns. Flawed or incomplete data can propagate errors, leading to incorrect findings or irreproducible results. The lack of human oversight in highly autonomous agents increases the risk of compounding errors, which can have serious consequences in fields like chemistry and biology, where precision and safety are critical. Furthermore, agent misalignment with research goals can lead to irrelevant or wasteful experiments, while multi-agent systems may suffer from coordination failures. In experimental automation, agents might deviate from established protocols or overlook key safety measures, potentially resulting in hazardous outcomes. As autonomy grows, the predictability and control of these agents must be carefully monitored to avoid unintended actions that are difficult to detect or correct in real time. Finally, the "blast radius" of these agents—especially those integrated with robotic labs—must be well-defined. Autonomous agents that interact with physical systems may misinterpret situational contexts, leading to unexpected escalations or system failures. Ensuring robust AI governance and human oversight is crucial for mitigating these risks, maintaining reliability, and reinforcing AI's role as a collaborative tool rather than an independent decision-maker in scientific research.

## 8 CONCLUSION AND FUTURE DIRECTIONS

Agentic AI for scientific discovery has shown inspiring results in domains such as chemistry, biology, materials science among others, attracting growing research interest. In this survey, we systematically review agentic AI approaches for scientific discovery by examining various aspects of its functional frameworks. Furthermore, we summarized its taxonomy, the important role of literature review in its workflow, and different approaches proposed in the recent years. By emphasizing widely used datasets and benchmarks, as well as addressing current challenges and open problems, we aim for this survey to serve as a valuable resource for researchers using agentic AI for scientific discovery. We hope it inspires further exploration into the potential of this research area and encourages future research endeavors.

Our analysis shows that while previous systems have performed well in fields such as chemistry, biology, and general science, literature review remains a significant challenge across nearly all approaches, especially in tasks like research idea generation(Baek et al., 2024) and scientific discovery (Schmidgall et al., 2025). For instance, Schmidgall et al. (2025) reported that among the phases of data preparation, experimentation, report writing, and research report generation, the literature review phase exhibited the highest failure rate. Similarly, while ResearchAgent is effective at generating novel research ideas, it lacks the capability to perform structured literature reviews, which are essential for grounding generated ideas in existing knowledge (Baek et al., 2024). The same limitation was observed in The AI Scientist framework (Lu et al., 2024). Another important future direction is the integration of calibration techniques into AI agents to improve the accuracy and reliability of their outputs in scientific discovery. Calibration ensures that the system's confidence in its predictions aligns with their actual correctness, which is critical in high-stakes domains such as healthcare. By incorporating these techniques, AI agents could become more trustworthy and effective tools for researchers, enhancing the reliability of their contributions to scientific research.

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
