# OpenReview forum: "Agentic AI for Scientific Discovery: A Survey of Progress, Challenges, and Future Directions"
_ICLR.cc/2025/Workshop/AgenticAI — ICLR 2025 Workshop AgenticAI Poster_

### Official Review · Reviewer_P6F5 · 2025-03-02
**a comprehensive survey**

**Rating:** 6
**Confidence:** 4

**Review:**

This survey provides a comprehensive overview of Agentic AI for scientific discovery, categorizing existing systems and tools, and highlighting recent progress across fields such as chemistry, biology, and material science. The paper also discussed key evaluation metrics, implementation frameworks, and commonly used datasets to offer a detailed understanding of the current state of the field.

Some parts of the paper are a little hard to follow. The authors can include more comparison tables/figures to assist the reader's understanding

---

### Official Review · Reviewer_zqmK · 2025-03-03
**Review of the Survey Paper**

**Rating:** 6
**Confidence:** 4

**Review:**

Summary:
This survey examines Agentic AI in scientific discovery, covering literature review automation, hypothesis generation, and experimentation. It reviews progress in chemistry and biology, discusses evaluation metrics, and addresses challenges like system reliability and ethics. Future directions focus on human-AI collaboration and improved system calibration for enhanced research effectiveness.

Strengths:
1) This paper provides a broad survey of existing Agentic AI systems, frameworks, and methodologies across multiple disciplines, including chemistry, biology, and materials science.
2) This paper outlines key implementation tools, datasets, and evaluation metrics, offering practical insights for researchers looking to develop or assess Agentic AI systems.
3) While highlighting the advancements of Agentic AI, this paper also critically discusses its limitations, such as challenges in automating literature reviews, concerns about system reliability, and ethical considerations. It emphasizes the importance of human-AI collaboration and system calibration to improve reliability.

Weaknesses:
1) This paper reviews various Agentic AI systems but does not provide a quantitative comparison of their effectiveness, scalability, or robustness. A benchmark study comparing different methods on common tasks would strengthen its contribution.
2) Although this paper discusses implementation tools and frameworks, it does not provide detailed guidance on how researchers can practically integrate Agentic AI into their workflows. More case studies or implementation examples would enhance its usability.
3) Figure 1 is referenced on page 4 but appears on page 6, making it difficult for readers to follow the discussion. Consider placing the figure closer to its first mention by using a long-format workflow graph.

---

### Decision · Program_Chairs · 2025-03-05

Accept (Poster)